# Effect of Fluorine Incorporation on DLC Films Deposited by Pulsed Cathodic Arc Deposition on Nitrile Butadiene Rubber and Polyurethane Rubber Substrates

**Lijie Zhang [1,2], Xuemei Zong [1,3], Fei Guo [3], Bing He [1,3] and Xiaoming Yuan [1,2,3,*]**

[1] Hebei Provincial Key Laboratory of Heavy Machinery Fluid Power Transmission and Control, Yanshan University, Qinhuangdao 066004, China; zhangljys@126.com (L.Z.); zxmxuzhou@163.com (X.Z.); hebing5280@163.com (B.H.)

[2] The State Key Laboratory of Fluid Power and Mechatronic Systems, Zhejiang University, Hangzhou 310058, China

[3] Jiangsu Xuzhou Construction Machinery Research Institute Co., Ltd., Xuzhou 221004, China; gf52284959@hotmail.com

[*] Correspondence: yuanxiaoming@ysu.edu.cn

**Abstract:** Diamond-like carbon (DLC) and fluorinated diamond-like carbon (F-DLC) films were deposited via pulsed cathodic arc deposition on pretreated nitrile butadiene rubber (NBR) and polyurethane (PU) rubber substrates. Both DLC and F-DLC films showed a more than 50% decrease in coefficients of friction compared to uncoated NBR and PU rubber substrates. The incorporation of fluorine was found to have little influence on the film coefficient of friction. However, a decrease in film wettability was overserved in the F-DLC films compared to the DLC films.

**Keywords:** F-DLC; pulsed cathodic arc; rubber; wettability; tribological performance

## 1. Introduction

Diamond-like carbon (DLC) films have been extensively studied due to their excellent properties: low coefficient of friction [1], anticorrosion [2], high hardness [3] and self-lubricating ability [4]. Owing to these characteristics, DLC films are widely applied in the surface coating industry, such as in cutting tools [5], hydraulic components [6], automotive components [7] and rubber seals [8]. Furthermore, the incorporation of other elements, such as F, Cr and Ti, can improve the properties of pure DLC films due to the alteration of film structures [9–11]. However, only a few studies have been reported regarding fluorinated diamond-like carbon (F-DLC) films deposited on rubber substrates [12,13], where the fluorine incorporation may alter the properties of DLC thin films. The incorporation of F content Rubber seals has been widely used in a number of engineering applications. However, one common failure mechanism of rubber seals is adhesive and abrasive wear due to high friction and rubber hysteresis [14]. With a thin layer of F-DLC film coated on the surface, the rubber component can be protected from wear and degradation.

F-DLC thin films are commonly created by means of chemical vapour deposition from hydrocarbon and perfluoroalkanes precursors [15–17]. Nobili and Guglielmini studied the thermal stability and mechanical properties of fluorinated diamond-like carbon coatings prepared by a plasma-assisted chemical vapour deposition method, and found that film hardness and fluorine content decreased with increasing annealing time at 500 °C [15]. Hasebe et al. studied the lubrication performance of F-DLC thin films deposited by radio frequency chemical vapour method [16]. The coated samples

exhibited an approximately 30% increase in lubricity compared with uncoated samples [16]. Yu et al. studied fluorinated amorphous diamond-like carbon film deposited by the plasma-enhanced chemical vapour deposition technique. The fraction of CF and $CF_2$ bonding configurations increased.

Another common method to deposit F-DLC thin films is physical sputtering, in which a target is bombarded by ionised atoms and target material atoms are ejected towards a substrate, forming a thin film. Wang et al. investigated the structures and tribological properties of F-DLC coating deposited on Ti-6Al-4V alloys using a hollow cathode plasma immersion ion implantation deposition system. The film hardness, Young's modulus and surface roughness decreased with an increase in F content in the films [18]. Imai et al. created F-DLC films using a T-shaped filtered arc deposition system and reported that film hardness decreased from 48 to 10 GPa as fluorine content increased from 6 to 22 atomic % [19]. Lin et al. also used a filtered cathodic vacuum arc system to deposit F-DLC films and found an increase in F content in the film with increasing Ar flow rate during deposition [20].

In this work, the pulsed cathodic arc method is employed for deposition of DLC and F-DLC thin films on rubber substrates. Comparing to the common cathodic arc deposition method or filtered cathodic arc deposition, the deposition temperature of the pulsed cathodic arc method is much lower (approximate 80 °C) and the number of marco-particles is kept at a minimum level. These unique features provide the possibility of thin film deposition on rubber substrates.

It is evident that both deposition techniques can create desired F-DLC films with various properties. Whilst there have been some advances in the properties of fluorine-containing DLC films [21], much less is known of pulsed cathodic arc deposited F-DLC films. In addition, the incorporation of F content can enhance the hydrophobicity of the DLC films, which can expand the application of DLC films to rubber seals in humid and marine environments. This study has investigated the effect of fluorine incorporation on DLC films deposited by pulsed cathodic arc deposition on nitrile butadiene rubber (NBR) and polyurethane (PU) rubber substrates.

## 2. Experimental Methods

The nitrile butadiene rubber and polyurethane rubber sheets were used as sample substrates with a size of 30 mm × 30 mm × 2 mm. The substrates were pretreated in tetrachloroethylene at 50 °C for 15 min in order to remove the plasticizer in the rubber and enhance adhesion between the film and rubber substrates [22]. After this step, both rubber substrates were dried in an oven at 150 °C for 1 h. Then, the substrates were mounted onto sample holders and placed into vacuum chamber. Both DLC and F-DLC films were deposited by means of the pulsed cathodic arc method. The cathodic target is a pure graphite target with a diameter of 29 mm. During deposition of the F-DLC films, Ar and $C_4F_8$ were introduced into the chamber with the same flow rate at 4 sccm for fluorine incorporation. The electrical arc was ignited under a voltage of 280 V with a pulse rate of 3 Hz on the surfaces of two carbon cathodic targets and the chamber pressure was kept at 0.05 Pa (0.1 Pa for F-DLC thin film) during the entire process (6000 pulses).

The as-deposited film thickness was measured using a Bruker Inc. Contour GT-K 3D optical microscope. The surface morphologies of as-deposited F-DLC thin films were observed by a FEI Inc. (MA, USA) Quanta FEG 250 scanning electron microscopy (SEM). The F content in the film was measured using a JEOL Inc. (Tokyo, Japan) 8530F Plus Electron Probe Microanalyser. Film Raman spectrum was obtained using a Thermo Fisher Scientific Inc. (MA, USA) DXR 2xi Raman Imaging Microscope with an excitation wavelength of 532 nm, laser power 10 mW and 2 s accumulated time. The tested samples for the Raman analysis were coated on the 10 mm × 20 mm × 0.5 mm (100) single crystal Si substrates.

The tribological performances of uncoated NBR and PU rubber samples, DLC film coated samples and F-DLC film coated samples were tested by a Bruker Inc. (MA, USA) UMT-2MT tribo-meter. All the samples were mounted onto the equipment and tested under a ball-on-disc dry friction scenario at 10 cm/s sliding speed with a 1 N load. The counterpart is a Ø10 mm 100Cr6 steel ball of hardness HRC 62. The surface roughness is measured by using a Bruker Dimension Icon Atomic Force Microscope

(AFM, MA, USA). The basic deposition parameters, film thickness and fluorine composition are listed in Table 1.

**Table 1.** Deposition parameters, film thickness and fluorine composition.

| Substrate | Voltage (V) | Ar/C$_4$F$_8$ Flow Rate (sccm) | Arc Frequency (Hz) | Thickness (nm) | F (at. %) | C (at. %) |
|---|---|---|---|---|---|---|
| NBR | 280 | 4/0 | 3 | 295 | 0 | 100 |
| NBR | 280 | 4/4 | 3 | 308 | 4 | 96 |
| PU rubber | 280 | 4/0 | 3 | 289 | 0 | 100 |
| PU rubber | 280 | 4/4 | 3 | 298 | 4 | 96 |

## 3. Results and Discussion

Figure 1 shows the SEM micrographs of DLC films and F-DLC films on NBR and PU rubber substrates. Figure 1a,b are SEM micrographs of DLC films on NBR and PU rubber, respectively, and Figure 1c,d are SEM micrographs of F-DLC films on NBR and PU rubber, respectively. Both DLC and F-DLC films exhibit a cracked surface structure with a segment size of 2–4 μm on NBR substrates. Meanwhile, same thin films show similar cracked segment morphologies with the segment size of less than 1 μm on PU rubber substrates. This unique surface condition is caused by thermal mismatch stress during film growing [22,23]. It is evident that incorporation of 4% F imposes little effect on film surface morphologies compared to the morphologic differences caused by the substrate materials.

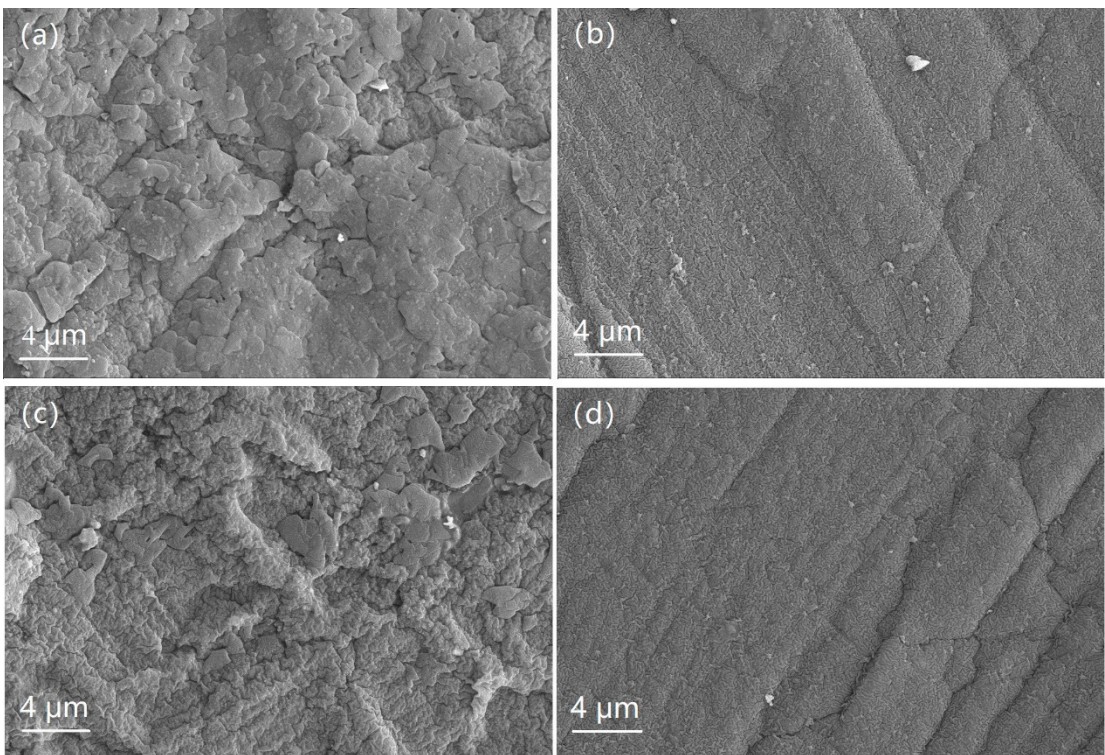

**Figure 1.** SEM micrographs of coated samples. (**a**) DLC film on NBR; (**b**) DLC film on PU rubber; (**c**) F-DLC film on NBR; (**d**) F-DLC film on PU rubber.

To investigate film structures, Raman spectroscopy was applied, which is a common, fast and non-destructive method to characterise carbon materials [24]. In the Raman spectrum, all carbon materials show similar features in the 1000–2000 cm$^{-1}$ region, where two distinct peaks can be observed at around 1560 and 1360 cm$^{-1}$, namely the G peak and D peak, respectively. The G peak is due to the bond stretching of all pairs of $sp^2$ atoms in both rings and chains. The D peak is due to the breathing modes of $sp^2$ atoms in rings [25]. Figure 2 presents Raman spectra of DLC and F-DLC films,

where the D peak and G peak are clearly observed and marked out by two vertical lines. Therefore, both as-deposited films can be characterised as typical DLC films. The $I_D/I_G$ ratios of DLC and F-DLC film are 0.46 and 0.45. The D peak and G peak positions are 1368, 1549 cm$^{-1}$ for deposited DLC films, and 1389, 1552 cm$^{-1}$ for F-DLC films, respectively. The full width at half maximum (FWHM) of the D peaks and G peaks are 333, 208 cm$^{-1}$ for deposited DLC films, and 269, 192 cm$^{-1}$ for F-DLC films, respectively. There is a slight shift to the higher wavenumber as the incorporation of F in the film for the D peak position whereas the G peak positions are approximately the same for both films. The similar $I_D/I_G$ ratios and G peak positions suggest that there is a similar amount of graphite clusters in both deposited films [26,27], which means that 4% F content does not impose a significant effect on the chemical structure of deposited thin films.

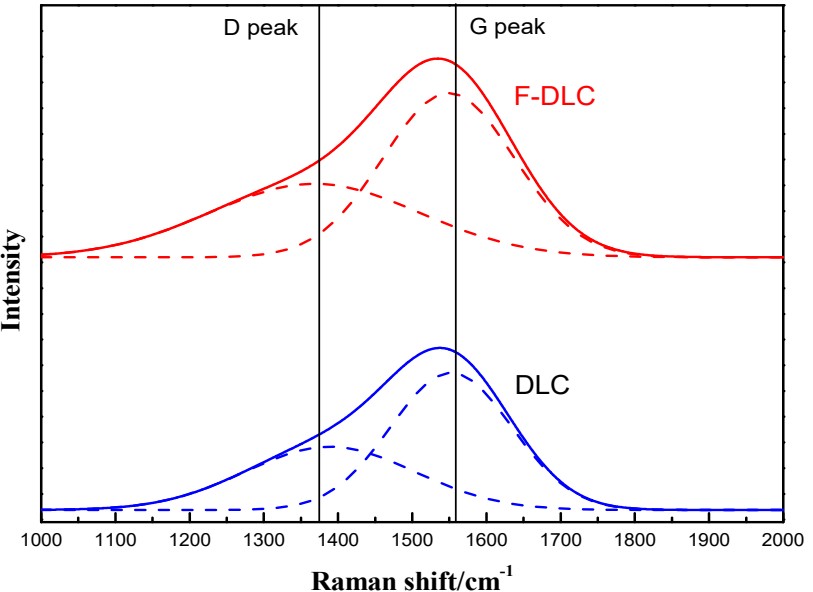

**Figure 2.** Raman spectra of DLC and F-DLC films.

　　The coefficients of friction (COFs) of uncoated NBR, PU rubber samples, DLC and F-DLC coated rubbers are shown in Figure 3. The coefficient of friction of NBR remains at approximately 1.2–1.4 during the tribological test, whereas both DLC and F-DLC coated NBR substrates present much lower COFs, approximately 0.2–0.35. As for PU rubber, the COF of the uncoated sample increases in the first 500 s and then remains stable at approximately 1.5. The DLC coated sample shows a similar trend for COF, where the COF increases from 0.2 to 0.6 during the first 500 s and stays at around 0.6 in the next 2500 s of the tribological test. The F-DLC thin film on PU rubber substrate exhibits a different trend compared to the DLC thin film on the same substrate. The COF of F-DLC thin film increases gradually, from approximately 0.3 to 0.6, over the entire tribological experiment. This phenomenon is discussed later in the next paragraph, where a close observation by SEM is carried out on the wear track morphologies. The improvements of tribological performance of DLC coated NBR and PU rubber substrate are expected and consistent with results reported in the literature [22,23,28]. The mass loss of the samples after the tribo-test were measured and displayed in Figure 3. It is evident that both uncoated NBR and PU rubber samples have approximately three times more mass loss than the coated DLC films on both substrates, which is consistent with the sample CoF results. The F-DLC film on the NBR substrate shows similar mass loss compared to the DLC film coated on the same substrate. However, the mass loss of F-DLC film coated on the PU rubber (4.56 mg) has a greater value compared to the DLC film coated on the PU (1.89 mg) rubber substrate.

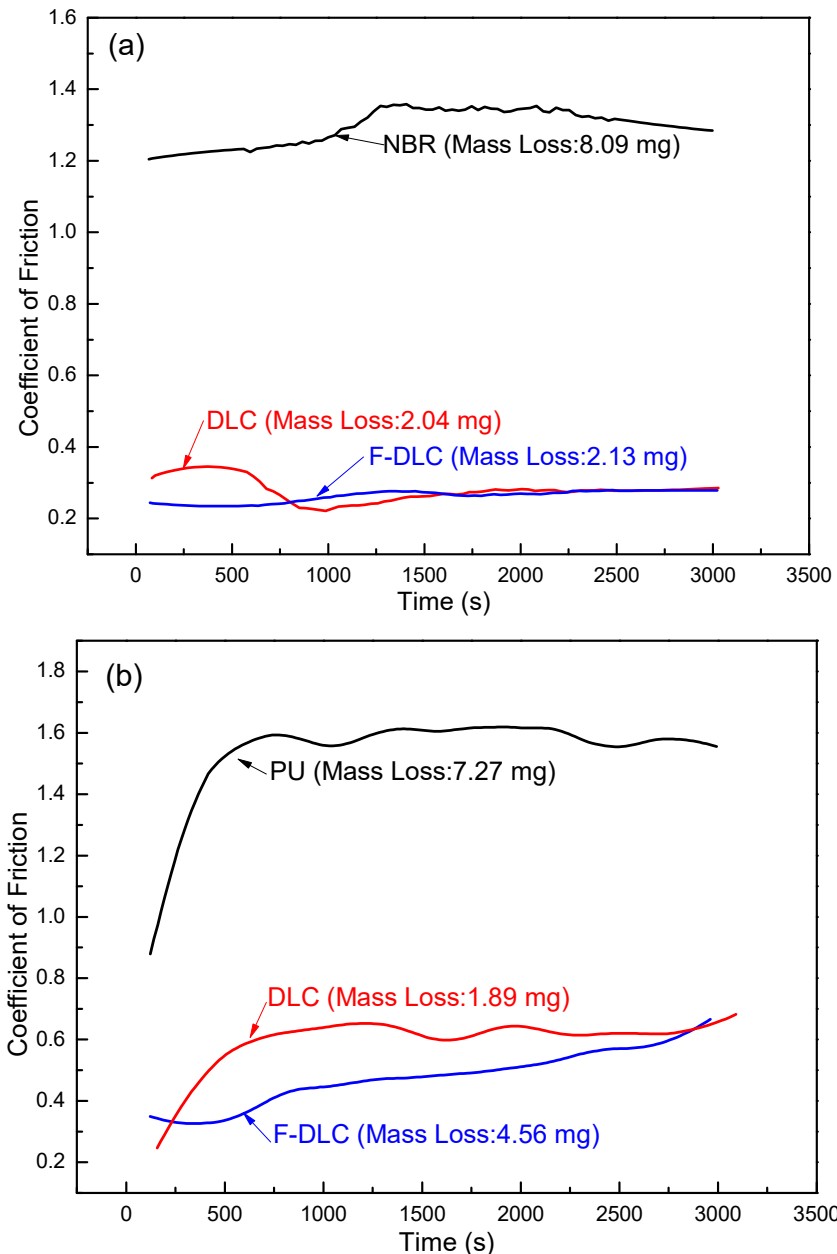

**Figure 3.** Coefficients of friction and film mass loss after tribo-tests of (**a**) uncoated nitrile butadiene rubber (NBR), deposited DLC and F-DLC films on UBR substrates; (**b**) uncoated PU rubber, deposited DLC and F-DLC films on PU rubber substrates.

Figure 4 shows three-dimensional AFM morphologies of uncoated NBR, PU rubber samples, and DLC, F-DLC films on both substrates as measured by AFM on a $5.0 \times 5.0$ μm$^2$ area. Figure 4a–c present uncoated and DLC, F-DLC film coated NBR samples. The surface of roughness ($R_a$) are 25.2, 41.7 and 41.8 nm, respectively. Figure 4d–f present uncoated and DLC, F-DLC film coated PU rubber samples. The surface of roughness ($R_a$) are 9.2, 16.7 and 17.3 nm, respectively. It is evident that the uncoated PU rubber and film coated PU rubber samples are smoother than NBR rubbers, which is consistent with the SEM results where the cracked segment size is larger on NBR substrates. In addition, the coated DLC and F-DLC films show higher surface roughness compared to both types of uncoated sample surfaces. This can be attributed to the cracked surface features as well.

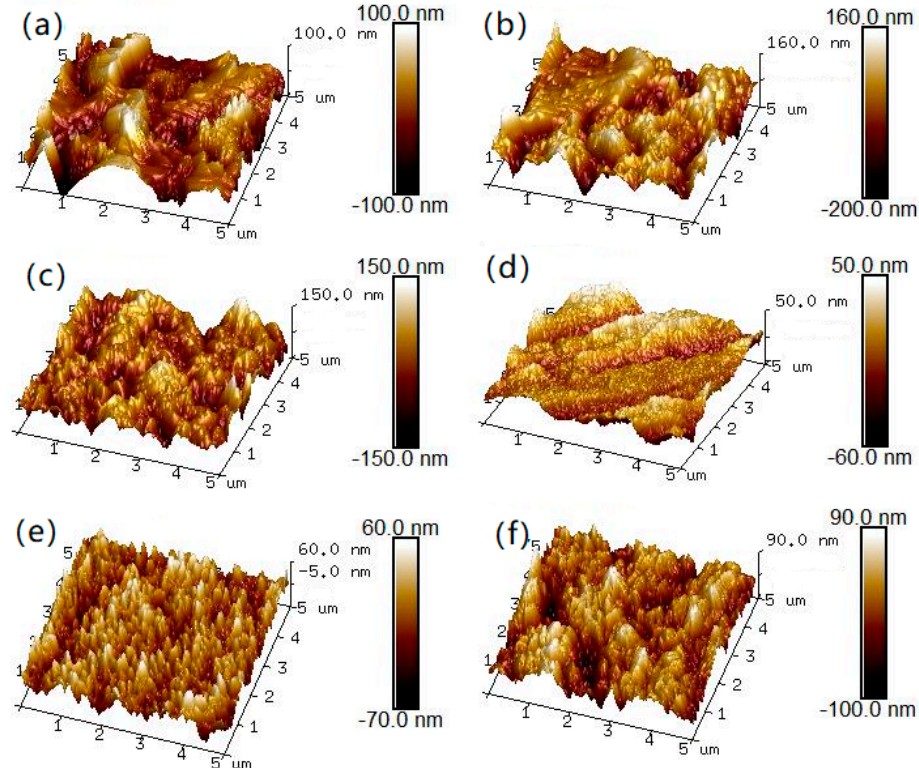

**Figure 4.** Three-dimensional AFM morphologies of uncoated substrates, DLC and F-DLC films: (**a**) uncoated NBR, (**b**) DLC on NBR, (**c**) F-DLC on NBR, (**d**) uncoated PU rubber, (**e**) DLC on PU rubber and (**f**) F-DLC on PU rubber.

The wear track morphologies are examined using a SEM microscopy, and the SEM micrographs are shown in Figure 5. Two SEM micrographs are obtained at 500× and 5000× magnifications for each sample. The wear tracks of the DLC and F-DLC coated NBR substrates are shown in Figure 5a,b,e,f, respectively. The wear tracks on these samples are hardly visible and the films remain intact after the tribological test in comparison to the as-deposited films presented in Figure 1a,c. However, the DLC and F-DLC films deposited on the PU rubber substrates present different friction tracks compared to the films deposited on the NBR substrates. Figure 5c,d display the wear track morphologies of the DLC film on the PU rubber substrate, where multiple film peeling-offs can be observed over the entire wear track area. Besides these peeling-offs, the rest of the film in the wear track region remains intact. The surface morphologies of F-DLC film on the PU rubber substrate after the tribo-test are shown in Figure 5g,h. The whole film has been peeled off, leaving adhesive wear tracks covering the entire friction region, which explains the abnormal mass loss of this sample presented in Figure 3. As shown in Figure 3b, the COF of this sample gradually increases during the tribo-test, which can be attributed to the F-DLC continually peeling off and forming adhesive wear tracks. The presence of these wear tracks and peeling offs leads to the observed high COF values for films deposited on the PU rubber substrates as well. Both DLC and F-DLC films show excellent tribological performance on NBR rubber. The incorporation of F content does not diminish the advantages of pure DLC films: wear resistance and low coefficient of friction on NBR rubber substrate. However, the deposited DLC film on PU rubber exhibits peeling-offs during the tribo-test, and the incorporation of F content worsens the peeling off, causing additional adhesive wear tracks. This may be attributed to the F content weakening adhesion between the film and the substrate.

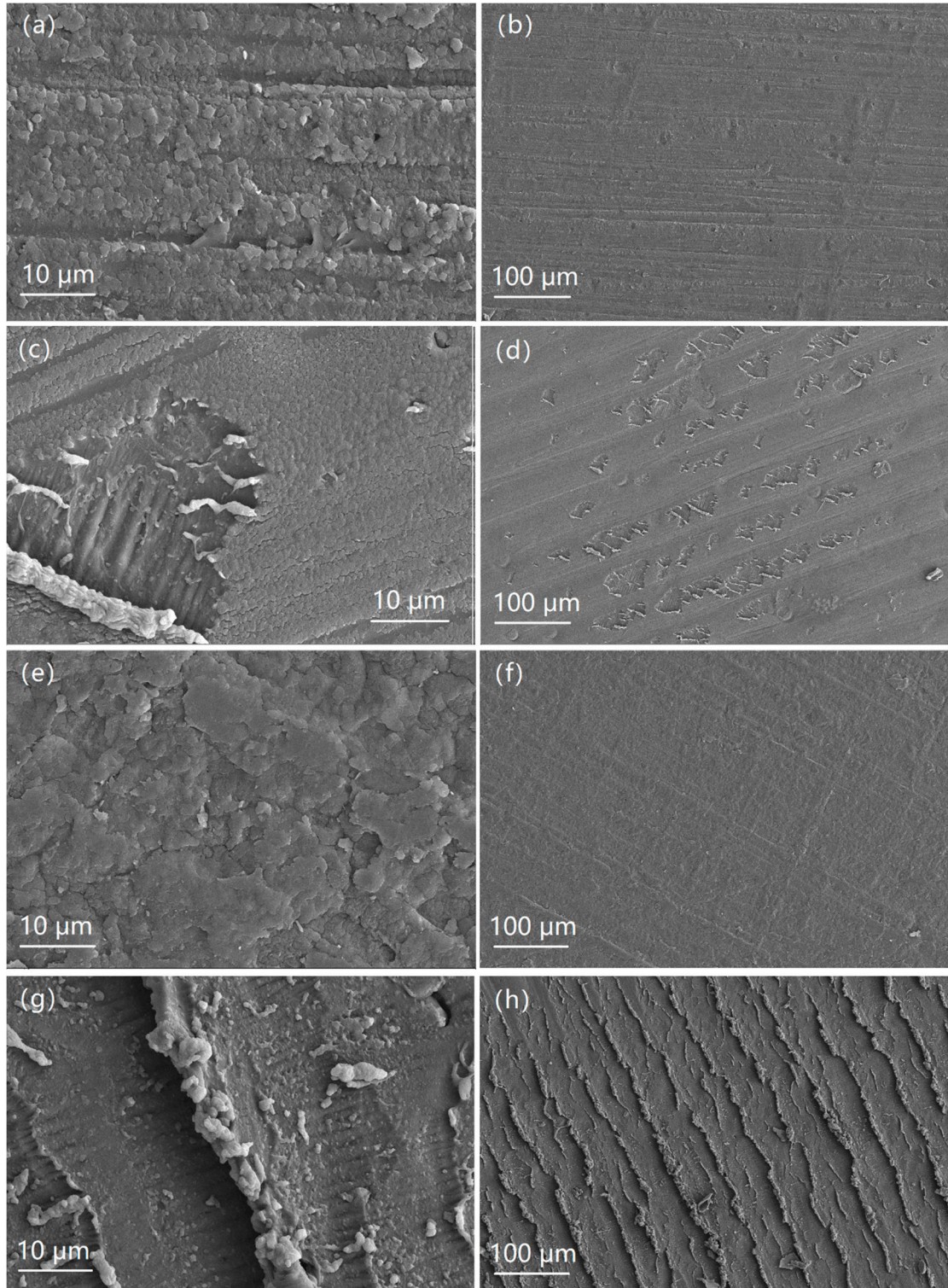

**Figure 5.** SEM micrographs of wear tracks for coated samples. (**a**,**b**) DLC film on NBR; (**c**,**d**) DLC film on PU rubber; (**e**,**f**) F-DLC film on NBR; (**g**,**h**) F-DLC film on PU rubber.

One common effect of introducing F content in the DLC thin film is the improvement of hydrophobicity [17,29,30]. All the contact angle results are listed in Table 2, and Figure 6 shows the contact angle test images of an uncoated PU rubber and a F-DLC film on PU rubber. The contact angle indicates the hydrophobicity of a solid surface. The hydrophobicity of deposited films is mainly affected by the incorporation of F content, surface roughness and the presence of $CF_n$ groups on the

surface [31]. The contact angle does not show a clear increase as F content introduced into the DLC films on NBR substrates. This is due to the nature of the cracked network features of the film surface. As presented in Figure 4, the surface roughness of thin films deposited on NBR substrates are much higher compared to the films deposited on the PU rubber substrates. Therefore, for the samples on the NBR substrates, the contact angle results are dominantly determined by surface roughness. On the contrary, for the F-DLC thin film deposited on PU rubber substrate, the contact angle exhibits an obvious increase as F content is presented, since the film surface roughness is much less compared to the films deposited on the NBR substrates.

**Table 2.** Contact angle results for all the samples.

| Material | Contact Angle (°) | Material | Contact Angle (°) |
|----------|-------------------|----------|-------------------|
| Uncoated NBR | 109.2 | Uncoated PU rubber | 89.3 |
| DLC on NBR | 110.9 | DLC on PU rubber | 98.8 |
| F-DLC on NBR | 112.3 | F-DLC on PU rubber | 117.5 |

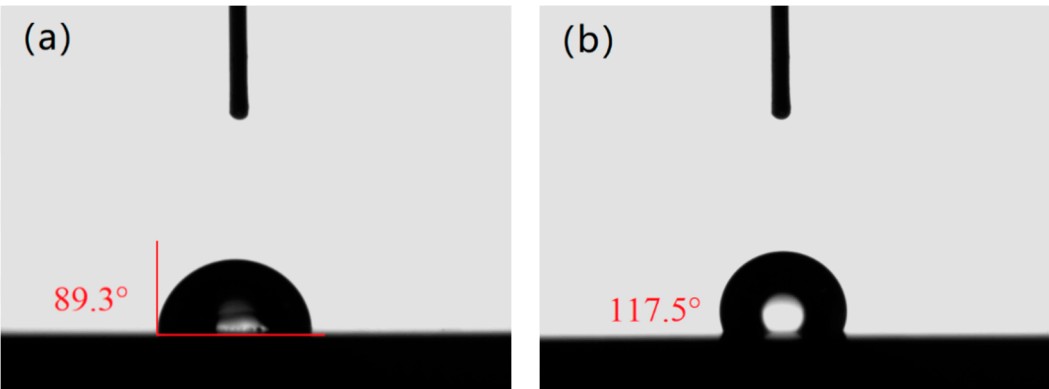

**Figure 6.** The contact angle images of (**a**) uncoated PU rubber; (**b**) F-DLC thin film on PU rubber substrate.

## 4. Conclusions

In this paper, DLC and F-DLC thin films have been deposited onto NBR and PU substrates by means of the pulsed cathodic arc method. The effect of incorporation of fluorine on film surface morphologies, chemical structures, surface roughness, tribological performance and hydrophobicity properties are investigated. The following conclusions are obtained:

- The surface morphologies of deposited DLC and F-DLC thin films exhibit cracked network features. The size of cracked segments shows a substrate dependence, where 2–4 μm segments are observed on the NBR substrate and less than 1 μm segments are found on the PU rubber substrate. This phenomenon is caused by thermal mismatch stress between the films and substrates. Raman spectroscopy is applied to all the deposited samples and typical DLC film features are characterized. Deposited DLC and F-DLC films have similar amounts of graphite cluster content in the film structure. The incorporation of fluorine does not present noticeable effects on film morphologies and chemical structures. In addition, the deposited films on PU rubber substrates have smoother surfaces compared to the films deposited on NBR substrates as a result of the cracked segment size.

- The tribological performances are tested under a dry sliding friction condition. It is evident that the COFs of deposited films are more than 60% lower than the uncoated substrates: a decrease from approximately 1.3 to 0.3 for NBR and from approximately 1.6 to 0.6 for PU rubber. The wear track morphologies are also studied by SEM microscopy. Both DLC and F-DLC thin film deposited on NBR substrates remain intact and no clear wear tracks are observed after the tribo-test. However, the films deposited on the PU rubber substrates show peeling offs and adhesive wear tracks after

the tribo-test and the incorporation of fluorine worsens the friction wear condition. The films deposited on the NBR substrates have higher surface roughness compared with the films deposited on the PU rubber substrates. The COFs of tested films present a contrary trend where the smoother films (films deposited on the PU rubber) have higher COF values. This is caused by the adhesive wears and peeling offs formed during the tribo-tests.

- The incorporation of fluorine improves film hydrophobicity for DLC films deposited on PU rubber substrate, where surface roughness does not play a dominant role in influencing film hydrophobicity properties, as is the case for films deposited on NBR substrates.

**Author Contributions:** Conceptualization, X.Z.; methodology, L.Z., X.Z., and F.G.; software, B.H. and X.Y.; validation, F.G. and B.H.; formal analysis, X.Z.; investigation, X.Z.; resources, F.G.; data curation, B.H.; writing—original draft preparation, X.Z.; writing—review and editing, F.G. and B.H.; visualization, X.Z.; supervision, L.Z. and X.Y.; project administration, L.Z. and F.G.; funding acquisition, L.Z., B.H., and X.Y. All authors have read and agreed to the published version of the manuscript.

**Funding:** This research was funded by the National Key Research and Development Project (2019YFB2005302), the Natural Science Foundation of Hebei Province (E2020203090), and Open Foundation of the State Key Laboratory of Fluid Power and Mechatronic Systems (GZKF-201820).

**Acknowledgments:** The authors are grateful for financial support from the National Key Research and Development Project (2019YFB2005302), the Natural Science Foundation of Hebei Province (E2020203090), and Open Foundation of the State Key Laboratory of Fluid Power and Mechatronic Systems (GZKF-201820).

**Conflicts of Interest:** The authors declare no conflict of interest.

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
