# Peer review of "Effect of Fluorine Incorporation on DLC Films Deposited by Pulsed Cathodic Arc Deposition on Nitrile Butadiene Rubber and Polyurethane Rubber Substrates"

_coatings, doi:10.3390/coatings10090878_

Round 1

Reviewer 1 Report

Fluorine incorporation on DLC films is an interesting area of study, especially on NBR and PU rubber substrates. This is a well written manuscript with clearly presented interesting results. However, there is lack of important information, especially regarding the role of certain properties such as roughness or adhesion on the tribological behaviour. I have the following comments, questions and suggestions for further improvement of the work.

  1. There is a clear effect of roughness on the friction results presented in the paper. Have you measured the roughness of uncoated and coated samples?
  2. The structural evolution of carbon coatings with the addition of F have not been elucidated. Please, complement the intro by discussing the influence of doping on the amorphous carbon structure of DLC coatings. Find some references of interest studying the influence of doping with Raman analysis:
    1. C. Trippe et al., Thin Solid Films 446 (2004) 85–90
    2. Jelinek et al., Applied Surface Science, Volume 417, 30 September 2017, Pages 73-83
    3. J.A. Santiago et al., Surface and Coatings Technology, Volume 382, 25 January 2020,124899
  1. There is clear enhanced tribological performance observed when DLC coatings are deposited. The formation of graphitic tribolayers could be the main factor on the reduction of friction coefficient. Do you have any evidence on the formation of lubritious tribolayers? Is the fluorine incorporation beneficial in this regard?

  1. The lack of adhesion figures along the wear tracks is an evident problem observed in SEM. Have you evaluated the wear rates? Do you have any measurement on residual stress of the coatings?

Reviewer 2 Report

The manuscript is interesting and contains a new information. However the additional measurements and comments should be added and more detailed explanation should be included.

  1. The used power, spot size and the accumulation time of Raman spectrometer should be included.

  1. Figure 2. presents the Raman spectra of DLC and F-DLC films. However it is not mentioned the substrate material. It is well know that the substrate material could affect the structure of the DLC films. So the authors should add the Raman spectra of both DLC and F-DLC films.
  2. Beside the ID/IG ratios, the D and G peak positions and FWHM of D and G peaks are important parameters and give an additional information about the structure of the deposited DLC films (see https://doi.org/10.1016/j.diamond.2019.107563, https://doi.org/10.1016/j.surfcoat.2018.05.012, https://doi.org/10.1016/j.tsf.2019.04.055, https://doi.org/10.1016/j.ceramint.2019.10.251). Those parameters should be included and analyzed in the manuscript.

4.The contact angle measurements indicated that DLC and F-DLC films deposited on different substrates had different values. Please add explanation about it. Usually it is related with the different structure of sp3 and sp2 carbon sites in the films.

  1. DLC and F-DLC films deposited on PU rubber substrates had a higher COF values compared to DLC and F-DLC films prepared on UBR substrates. Please indicate the reason of such results.
  2. Lines 152-153. The authors wrote: "The contact angle results are dominantly determined by surface roughness on these samples". However the information about the surface roughness of the used substrates and deposited coatings is missing. The surface roughness measurements (AFM or profilometry) should be included in the paper.
  3. The COF value is very important in the tribological application. The wear rate value is another parameter which is even more important in order to use coatings in the tribology. However the authors do not measured the wear rate of the coatings. Maybe the formation of the DLC films does not improved the wear rate due to insignificant adhesion between the substrates and the coatings?
  4. The conclusion should be corrected. The most important information should be included and the obtained results should be proved in the text with the additional measurements.
  5. 97-106 lines. The authors should check the indexes.
  6. Part of results was presented in Coatings202010(6), 545; https://doi.org/10.3390/coatings10060545.
  7. The authors indicated that the content of F in the films was 4 %. What was the carbon and oxygen content in the films? Add more information about the determination of chemical composition of the films.

Reviewer 3 Report

The work presented in this work deals with DLCs prepared by pulsed cathodic arc with and without fluorine incorporation on NBR and PU rubber substrates. The structure, flow and language quality of the presented work is good. The results have been presented clearly and the conclusions drawn seem appropriate. However, there are few important things that look missing or they could have been presented in a better way.

  • The first and foremost is the deposition technique used: in the introductory paragraphs, the authors review the literature and mention various deposition methods for preparing DLCs for such an application. If we only focus on cathodic arc based works, the authors outline few issues such as decrease of hardness, increase in fluorine content etc. Then what is the magic that the pulsed cathodic arc can do to solve this issue? How do the film properties presented in this work compare with another pulsed cathodic arc based work or other arc based works? This is clearly missing and therefore I strongly recommend the authors to present a comparison and highlight as to why the pulsed cathodic arc method is better than other arc based methods.
  • My other concern is with the motivation that is presented for this work. In the last paragraph of the introduction, the authors motivate the work by stating that much less is known for F-DLC with pulsed cathodic arc and it appears that this is the main motivation behind carrying out this research work. This does not look good enough motivation especially after looking at the results which clearly indicate that incorporation of fluorine doesn’t play a significant role in the tribological and morphological properties, rather it is the DLC film itself that leads to improvements as compared to uncoated substrates. I strongly recommend the authors to re-think and clearly state the motivation of this work showing the big picture and showing the reader the importance and relevance of this work in solving an existing problem.

  • In the experimental details section, it is mentioned that the substrates were pre-treated at 50 C and dried at 150 C for 1 hour. What maximum temperature these substrates can take? and is there any influence of the time for drying the substrates on the morphological and tribological properties?
  • It appears that 4% fluorine was used in the gas atmosphere, was this optimum fluorine content? What happens to film properties when the fluorine content is for example 10%? Or 1%? It will be good for the understanding if the authors can present an evolution of for example COF against the fluorine content.
  • What is the finding of the Raman based characterization? It is somewhat stated in the conclusions but missing in the results and discussion section. Please state clearly why analyzing the structure of the films using Raman was important and how the Raman results help build the case.
